# Oxidative Stress and Chemoradiation-Induced Oral Mucositis: A Scoping Review of In Vitro, In Vivo and Clinical Studies

**DOI:** 10.3390/ijms23094863

**Published:** 2022-04-27

**Authors:** Huynh Nguyen, Simran Sangha, Michelle Pan, Dong Ha Shin, Hayoung Park, Ali I. Mohammed, Nicola Cirillo

**Affiliations:** Melbourne Dental School, The University of Melbourne, Carlton, VIC 3053, Australia; huynhn1@student.unimelb.edu.au (H.N.); sksangha@student.unimelb.edu.au (S.S.); mpan2@student.unimelb.edu.au (M.P.); neil.shin@student.unimelb.edu.au (D.H.S.); hayoungp@student.unimelb.edu.au (H.P.); mohammeda2@student.unimelb.edu.au (A.I.M.)

**Keywords:** oral mucositis, chemoradiation, oxidative stress, reactive oxygen species

## Abstract

Chemoradiation-induced mucositis is a debilitating condition of the gastrointestinal tract eventuating from antineoplastic treatment. It is believed to occur primarily due to oxidative stress mechanisms, which generate Reactive Oxygen Species (ROS). The aim of this scoping review was to assess the role of oxidative stress in the development of Oral Mucositis (OM). Studies from the literature, published in MEDLINE and SCOPUS, that evaluated the oxidative stress pathways or antioxidant interventions for OM, were retrieved to elucidate the current understanding of their relationship. Studies failing inclusion criteria were excluded, and those suitable underwent data extraction, using a predefined data extraction table. Eighty-nine articles fulfilled criteria, and these were sub-stratified into models of study (in vitro, in vivo, or clinical) for evaluation. Thirty-five *clinical* studies evaluated antioxidant interventions on OM’s severity, duration, and pain, amongst other attributes. A number of clinical studies sought to elucidate the protective or therapeutic effects of compounds that had been pre-determined to have antioxidant properties, without directly assessing oxidative stress parameters (these were deemed “indirect evidence”). Forty-seven in vivo studies assessed the capacity of various compounds to prevent OM. Findings were mostly consistent, reporting reduced OM severity associated with a reduction in ROS, malondialdehyde (MDA), myeloperoxidase (MPO), but higher glutathione (GSH) and superoxide dismutase (SOD) activity or expression. Twenty-one in vitro studies assessed potential OM therapeutic interventions. The majority demonstrated successful a reduction in ROS, and in select studies, secondary molecules were assessed to identify the mechanism. In summary, this review highlighted numerous oxidative stress pathways involved in OM pathogenesis, which may inform the development of novel therapeutic targets.

## 1. Introduction

Oral mucositis (OM) is an acute inflammatory, ulcerative condition of the oral mucosa that commonly arises as a consequence of chemo- and/or radiotherapy. Treatments to manage cancer, such as radiotherapy, cisplatin or 5-fluorouracil (5-FU), generate ROS that target neoplastic cell DNA resulting in cell damage and death. ROS from these therapies also indiscriminately target healthy non-neoplastic DNA [1].

Production of ROS in excess leads to oxidative stress. Maintenance and regulation of ROS involves multiple enzymes with antioxidant properties including superoxide dismutase, catalase and glutathione peroxidase. The offset of this cell redox balance can be a result of oxidants that are formed following free-radical generation, or from a dysfunction in the antioxidant protective mechanism. The increase in oxidative stress is often associated with greater malondialdehyde and 4-hydroxynonenal, greater DNA damage, and protein structural impairment [2]. Biologically, oxidative stress can increase the risk of cancer and predisposition to inflammatory disease and conditions, among other things.

The resultant activation of the nuclear factor kappa B (NFκB) during chemoradiation causes an increase in pro-inflammatory cytokine production. This cascade of inflammatory pathways culminates in mucosal ulceration resulting in the clinical presentation of treatment-induced OM [3,4]. Patients developing OM are often debilitated by the inflammation and ulceration, affecting diet, appetite, and ability to conduct normal oral hygiene [5]. Furthermore, ulceration into the submucosa predisposes treated patients to secondary infections, causing a potential for further sequelae.

Chemo-/radiotherapy-induced ROS are implicated in OM, and therefore the oxidative stress pathway has been evaluated for therapeutic effect, including OM prophylaxis. However, it is yet to be elucidated if there is a specific mechanism in the oxidative stress pathway that can be modulated to prevent OM.

Glutathione (GSH), superoxide dismutase (SOD), catalase, myeloperoxidase (MPO), and hydrogen peroxide are some enzymes and compounds that are involved in the complex oxidative stress pathway [2]. This pathway also signals for the regulation of pro-apoptotic pathways such as NFκB, adding to further complication, but also potential therapeutic targets for OM prophylaxis.

Studies investigating oxidative stress and the role of antioxidants, in the context of chemo-/radiotherapy-induced OM, have assessed the effects of the improvement of OM in various settings. Various agents in these studies have been evaluated by assessing different enzymes of the oxidative stress pathway. This scoping review aims to collect the data generated from these studies to better understand the mechanisms by which the antioxidants can provide OM relief and improvement.

## 2. Results

There were 717 records identified in SCOPUS and PubMed, respectively. Out of these studies, 89 met the inclusion criteria and were included in the qualitative synthesis (Figure 1). A detailed description of each individual study is reported in Appendix A. The main characteristics of the studies included are described in the manuscript.

### 2.1. In Vitro Studies

There were 21 in vitro studies that were suitable for inclusion. Cell lines cultured in the studies were sheep red blood cells [6], marrow cultures from mice [7], human dermal fibroblast cells [8], keratinocytes [9,10,11,12,13,14,15,16,17,18,19,20], human lung [21] and periodontal ligament fibroblasts [14,22], gingival cells [23], epithelial cells [24] and pharyngeal cells [25,26] (Table 1).

Studies included potential novel therapeutic agents for OM. ROS scavenging activity was demonstrated by 5 studies, which investigated GS nitroxide JP4039, astaxanthin, green tea (polyphenol), amifostine and cyclooxygenase-1 inhibitor [7,8,9,14]. Decreased ROS production was demonstrated by 11 studies, which investigated rapamycin, rebamipide, novel compound 3-amino-3-(4-fluoro-phenyl)-1H-quinoline-2,4-dione (KR22332), Korean ginseng, Salvia miltiorrhiza Bunge (SM), Onchumg eun (OCE), epicatechin, γ-tocotrienol, Daiokanzoto (TJ84), N-acetyl cysteine (NAC) and photomodulation [10,11,12,13,15,16,17,18,20,25,26]. Protection against generated ROS was demonstrated by 5 studies: NAC and Qingre Liyan, azelastine, mucosamin (hyaluronic acid-based compound), phenylbutyrate, GS nitroxide JP4-039 [6,13,19,21,24].

### 2.2. In Vivo Studies

There were 47 in vivo studies investigating the capacity for various compounds for prophylaxis and reductive effects on OM severity and onset (Table 2). The studies retrieved for analysis have been categorized by intervention types, as reported below.

#### 2.2.1. Chemical Compounds

Sixteen studies assessed the use of different chemicals in the prevention of OM. These drugs provided prophylactic therapy for OM by preventing onset [10,17] or reducing severity [27,28,29,30,31,32,33]. Histologically, limited or absence of oral epithelial destruction, reduced cell apoptosis, vasodilation, and inflammatory cell infiltrates were reported [10,29,30,31,32,33]. Ulcerated epithelium had improved recovery and thickness when administered the drugs [31]. Protection of or reduction in OM were associated with reduced ROS [15,17,34]. Compared to negative control groups, greater OM severity was associated with reduced γH2AX [9,31,34], MDA [30,35,36], and MPO [29,30] expression; additionally, there was increased GSH [37,38] and SOD expression [10].

#### 2.2.2. Rebamipide

Three studies reported rebamipide exerting therapeutic effect by reducing ulcer-like lesions. The healing effect is suggested to be temporal-, frequency- and dose-dependent [39,40,41]. Rebamipide was associated with reduced pro-inflammatory protein expression [40,41].

#### 2.2.3. Botanical Extracts

Thirteen studies demonstrated the protective effects of botanical extracts against OM. The studies assessed histological and oxidative stress biomarkers such as ROS production, MDA, MPO and SOD [26,42,43,44,45,46]. Treatments with plant extracts led to a significant reduction in severity of OM [47], and accelerated recovery of the epithelial layers [25,26,48,49,50,51].

#### 2.2.4. MnBuOE

Two studies reported the protective effects of MnBuOE, a redox-active manganese porphyrin, against OM. MnBuOE treatment increased the GSH/GSSG ratio compared to the irradiated control group [52] and led to significant reduction in severity of radiation-induced OM [53].

#### 2.2.5. Laser Therapy

Laser therapy, in two studies, reduced OM severity, most significantly in extra-oral laser irradiation. Furthermore, laser therapy improved the rate of healing of OM and OM score; these were associated with reduced ROS generation, and increased glutathione peroxidase (Glu.Px) and SOD activity [54,55].

#### 2.2.6. Transplantation

A single study investigated the transplantation of CXCR2-overexpressing mesenchymal stem cells in mice following irradiation, reported improved OM healing. The macroscopic effect was associated with a reduction in ROS production [56].

#### 2.2.7. Antioxidants

Six studies reported the protective effects of antioxidants against OM. The antioxidant effects were evaluated by assessing various oxidative stress biomarkers and histological changes. Antioxidant treatment resulted in reduced severity and area of OM [7,24,57,58,59,60], and greater epithelial thickness [58,59], as compared to the control. Moreover, it led to later onset of OM compared to the control group [60]. In treatment groups with antioxidants, higher levels of SOD and catalase and lower levels of MDA were described, which illustrated decreased oxidative stress [60].

#### 2.2.8. Genes

***Gene therapy.*** Three studies investigated gene therapy in the context of OM. They reported reduced ulcer area, cell apoptosis, and associated reduced expression of DNA damage markers (pH2AX and 8-OhdG) in Tat-Smad7 treated mice following radiotherapy [61]. Two papers examined therapeutic potential of SOD plasmids in OM therapy following radiation insult. Both reported greater SOD expression, by SOD plasmid treatment, resulted in improvement in OM [62,63].

***MicroRNA.*** A single paper explored the modulatory effect of miR-200c in the pathogenesis of radiotherapy-induced oral mucositis (RIOM) in irradiated mice. The study demonstrated an increase in miR-141, miR-200a, miR-200b, and miR-200c expressions in irradiated mice that had developed OM [64].

### 2.3. Clinical Studies

Thirty-five clinical studies were included, some of which evaluated various interventions with antioxidant capabilities, whilst a small number of publications explored inherent protective factors in clinical subjects, such as genetic polymorphisms, hematological variations, and natively found constituents with antioxidant capabilities (Table 3).

#### 2.3.1. Vitamin E

There were eight studies that evaluated vitamin E’s (VE) capacity as a treatment or prophylactic measure for OM. Four studies identified the effect of topical VE independently, in which three showed a significant effect of VE on OM improvement, whilst one did not [65,66,67,68].

Three studies evaluated VE in conjunction with other interventions [6,69,70]. Two of those reported less severe grades of OM by VE [6,70]. The other study concurred with these findings, after adjusting for age [69]. One study compared Vitamin E’s efficacy to Pycnogenol (Pine Bark Extract) [71]. This reported that both interventions were equally effective in reducing the severity of OM and its associated pain.

#### 2.3.2. Genetic Influences & Inherent Antioxidants

Two studies evaluated the genetic link to OM, in particular genetic polymorphisms and their association with OM development. Both studies reported that particular polymorphisms (e.g., in XRCC1 or variants of NBN) are associated with a higher OM risk [72,73].

Two studies sought to correlate OM levels and plasma and buccal mucosa antioxidants levels. These studies found no correlation between antioxidant levels and OM severity [74,75].

One study investigated the effect of salivary antioxidants against OM [76]. They reported an increase in SOD levels coinciding with the development of OM, and a decrease in uric acid (UA) levels reflecting the progression of tissue damage. An association between severe acute OM and specific leukocyte lymphocyte, and plasma antioxidative capacity concentrations was also revealed [77].

#### 2.3.3. Rebamipide

Two clinical studies that investigated Rebamipide as a treatment for OM, and both reported that rebamipide results in decreased severity of OM [78,79].

#### 2.3.4. Zinc/Polaprezinc

Two clinical studies that investigated the effects of zinc supplementation on OM were included. Both demonstrated a decrease in the incidence of OM. One revealed that zinc promoted OM recovery [80,81].

#### 2.3.5. Selenium

There were two studies that evaluated selenium. One determined that supplementation with selenium did not affect OM, and the efficacy of radiotherapy [82]. The other determined that selenium resulted in reduction in severe OM incidence, but not in the cumulative incidence [83].

#### 2.3.6. Photobiomodulation

One study explored the modulatory effect of photobiomodulation on oxidative stress [20], specifically ROS reduction and antioxidant activity at different wavelengths. 800 nm laser light or a combination of 660, 800 and 970 nm light was discovered to result in the largest ROS reduction.

#### 2.3.7. Hyaluronic Acid-Based Compounds

Two clinical studies assessed the efficacy of hyaluronic acid-based compounds. Patients treated with hyaluronic acid-based compounds have been reported to show reduced incidence and pain [84]. Used as OM prophylaxis, the compounds have demonstrated minimal intensity or no recurrence when applied three times daily via oral spray [19].

#### 2.3.8. Single Study Per Intervention

For the following, only one study per intervention met inclusion criteria: NAC, propolis, genistein, glutamine, MF 5232, melatonin, actovegin, date palm pollen, β-carotene, GC4419, *Calendula*, allopurinol and erythropoietin (Table 4).

## 3. Discussion

The aim of this scoping review was to systematically appraise the relevance of oxidative stress pathways in the pathogenesis of radio-/chemotherapy-induced OM. We used a comprehensive approach by including studies undertaken in all experimental settings (in vitro, in vivo, and clinical studies) and providing direct or indirect evidence for a role of oxidative stress or antioxidants in the development, prevention or treatment of OM. In total, 89 papers were included, and these were sub-stratified into models of study (in vitro, in vivo, or clinical) for evaluation. There were 22 in vitro studies, 47 in vivo studies, and 35 clinical studies. Discrepancies in the study count are owed to the fact that some studies had numerous intervention arms in different populations, allowing them to be evaluated in triplicate.

Whilst all papers evaluated OM and involved some element of antioxidant mechanisms, only some could provide direct evidence for the role of this pathway.

### 3.1. Direct Evidence

#### 3.1.1. In Vitro

Most in vitro studies assessed the levels of ROS production by cells, providing direct evidence for the oxidative impact of each intervention at a cellular level. However, their connection to OM may yet remain unclear until in vivo or clinical trials are evaluated.

While demonstrating ROS reduction, the oxidative pathway, resulting in decreased oxidative stress, was further examined in some studies. Astaxanthin inhibited the cisplatin-induced release of intracellular ROS and inhibited human dermal fibroblast proliferation via peroxidation of the cytoplasmic lipids [8]. Rapamycin also demonstrated protection against senescence from DNA damage following H_2_O_2_ treatment, indicating it inhibits mTOR to suppress oxidative stress and reduce ROS accumulation [10]. KR22332 inhibited apoptosis-related genes such as p53, and TNF-α, suggesting its ROS protective capacity is related to TNF-α inhibition [11]. Korean red ginseng reduced ROS by stabilizing the change in MMP [12]. SM and OCE reduced intracellular ROS production and radical DPPH, demonstrating its antioxidant activity [17]. γ-tocotrienol and NAC advanced the accumulation of Nrf2, a transcriptional factor for cytoprotective gene, suggesting its protective effect during oxidative stress [18].

#### 3.1.2. In Vivo

Botanical extracts, MnBuOE, chemical compounds, plasmid treatment, CXCR2 overexpression, and vitamin E and L-carnitine interventions were reported to act on the oxidative stress pathway similarly in vivo. These studies demonstrated that higher SOD [10,26,42,43,44,45,60,62], and GSH [52,53] levels and lower MPO and MDA levels were associated with prevention or reduced severity of OM [10,26,29,30,35,36,42,43,44,45,49,51,52,53,60]. Interestingly, in one study, catalase, an enzyme of the oxidative stress pathway that exerts antioxidant function, was reduced following cannabidiol treatment [49]. As non-treated irradiated mice had higher catalase levels, it is possible that ROS production was mitigated by cannabidiol treatment, and therefore the regulation of oxidative stress was not required. However, the nature of catalase in the study is unclear due to a lack of direct ROS assessment.

Further limitations of the in vivo studies are the inconsistency of verifying the SOD expression in the plasmid therapy [62,63] and limited evidence correlating signs of OM improvement with ROS [17,26,34,36,56], a more direct method of measuring oxidative stress.

#### 3.1.3. Clinical Studies

The majority of clinical studies did not provide direct evidence as to how the interventions influenced the oxidative stress pathway. A small number of studies examined their antioxidant capacity, or inherent antioxidant pathways. We interpreted these studies as providing direct evidence.

Both photobiomodulation and *Calendula* provided direct evidence supporting their role in the oxidative stress pathway [20,97]. Photobiomodulation was shown to be associated with ROS generation due to its ability to excite cytochrome C oxidase at high wavelengths. However, the research in this field is limited, and a better understanding of how light interacts with biological tissues is necessary before adopting this intervention. Antioxidant activities were present in the *Calendula* intervention, including free radical scavenging and termination [97]. However, the literature claimed that a better understanding of optimal doses and administration frequency would better improve OM outcomes.

The study evaluating NAC provided some limited direct evidence for the role of the oxidative stress pathways and the development of OM. NAC stimulates the synthesis of GSH, which can scavenge free radicals. This study demonstrated that supplementation with parenteral NAC reduced the severity and duration of OM, and that serum levels of Glu.Px were higher in the intervention group—this implies that the antioxidant capacity of the compound is responsible for the changes in disease character [91].

Two studies evaluated OM’s association with particular genetic polymorphisms, providing direct evidence. Both studies assessed genes related to DNA damage and repair, particularly genes involved in protection against oxidative stress. Both studies concluded that variations in these genes are associated with a higher risk of developing OM [72,73]. This lends credibility to the assertion that the pathogenesis of OM is intrinsically linked to the oxidative stress pathway.

Three additional studies investigated the correlation between endogenous levels of antioxidants and OM. One study demonstrated that the natural salivary antioxidants, SOD, and uric acid levels were altered during OM progression [76]. Surprisingly, two studies reported no correlation between measures of plasma antioxidants and severity of OM, and that there was no particular antioxidant that had a predictive effect for severity or incidence of OM [74,75]. While this may seem to contradict the crux of the body of research, a more reasonable interpretation may be that there may be additional factors at play that affect the relationship between the oxidative stress pathway and OM. One finding from these studies that seems to support the rest of the available evidence is that there was a tendency for patients with sub-normal antioxidant concentrations to require a longer duration of parenteral nutrition, implying that their OM was symptomatic for longer. An interesting postulate that can also arise from this data is that perhaps the presence of antioxidant mechanisms is not protective, but the absence of sufficient antioxidant capacity is deleterious and could be the reason for the lack of protection.

A study on melatonin, a potent antioxidant, demonstrated its association with increased total antioxidant capacity (TAC) in patients with OM [94]. It further demonstrated that supplemented patients had reduced discomfort and pain. These results imply that the increased TAC is associated with improved OM outcomes.

Two studies about selenium provided direct evidence of its antioxidant effects. Increases in selenium levels were associated with an increase in GSH peroxidase levels which are important for endogenous detoxification of free radicals [82,83]. This was the assumed method by which selenium decreased OM severity.

### 3.2. Indirect Evidence

A number of clinical studies sought to elucidate the protective or therapeutic effects of compounds that have been pre-determined to have antioxidant properties. These include studies of propolis, genistein, glutamine, MF 5232, actovegin, vitamin E, date palm pollen, mucosamin (hyaluronic acid-based compound), rebamipide, mucosyte^®^, β-carotene, zinc, GC-4419, allopurinol and erythropoietin [6,19,65,66,68,69,70,71,78,79,80,81,84,85,86,87,88,89,90,92,93,95,96,97]. Whilst these studies provided evidence for the efficacy of these compounds, they did not explore the mechanisms behind how they produced an improvement in OM outcomes. Therefore, they only provided indirect evidence towards our research outcomes.

In vivo studies that demonstrated a reduction in ROS levels without assessing the oxidative stress mechanisms involved GS nitroxide JP05429, green tea (polyphenol), dexpanthenol, amifostine and cyclooxygenase-1-inhibitor (allopurinol), Daikanzoto, Hangeshashinto, rebamipide, L. reuteri, Z. jujuba, propolis, curcumin, velafermin, DMSO, GGsTop, antimicrobial laser or radiation therapy, Tat-Smad7 treatment, and tempol [7,9,14,22,23,27,31,33,39,40,41,42,47,48,50,54,55,57,58,59,61].

In vitro, phenylbutyrate [21], NAC and qingre liyan [13], and mucosamin [19] a direct or an indirect increase in ROS as indicated via changes in expression of particular proteins, such as heme-oxygenase-1 and mTOR [13,16]; or fibroblast senescence [19], suggested to be a result of reduced oxidative stress by free radical sequestration [19,20].

### 3.3. Limitations

There were a number of limitations within this scoping review. Whilst there has been a significant return of results, due to the immense heterogeneity, it is difficult to evaluate the quality of evidence. We have attempted to do so where we have found disagreement amongst the literature, but this is a primitive attempt. Furthermore, large differences in study populations and outcomes measured have prevented us from coming to a representative conclusion regarding our objective of determining the function of oxidative stress pathways in the development of OM. Due to constraints arising from the sheer amount of data retrieved, our scoping review made no attempt to evaluate the bias of the studies. However, since its inception, this study was planned to examine the breadth of the evidence rather than the depth, and these limitations are typical of scoping reviews. Future systematic reviews will build on the trends highlighted by our work and will focus on specific aspects on the oxidative stress pathway in OM.

## 4. Materials and Methods

### 4.1. Search Strategy

The scoping review was conducted in accordance with PRISMA-ScR guidelines [98].

The following databases were searched:PubMed (MEDLINE)SCOPUS

Inclusion criteria were studies on OM in patients receiving chemo-/radiotherapy, in vivo or in vitro models of OM, assessment of oxidative stress or related pathways and original articles (including case-reports). Gray literature was not considered in this review.

No restrictions were placed on the date of publication when searching databases. Publications released prior to the month of June 2021 were considered in this review.

The aim of this scoping review was to understand the mechanism of the oxidative stress pathway in chemo-/radiotherapy-induced OM. To initially screen a broad range of studies, synonyms for OM were considered in the search strategy. Furthermore, oxidative stress has recently been of interest in the prevention of OM. Therefore, the oxidative stress pathway was considered in the search strategy. Moreover, enzymes commonly studied in the oxidative stress pathway and terms associated with oxidative stress were included in the search strategy.

From this, the search strategy included three concepts:The condition: OM;Aetiology of the condition: chemotherapy and radiotherapy;Pathogenesis/pathway of interest: oxidative stress pathway.

The following keyword combinations were used to find relevant publications: (chemotherapy * OR “radiation-induced” OR radiotherapy *) AND (mucositis OR “Oral mucositis” OR “mucosal atrophy” OR “alimentary tract mucositis” OR “gastrointestinal tract mucositis” OR stomatitis) AND (“Reactive Oxygen Species” OR “oxidat *” OR “free radicals” OR ROS OR “Superoxide Dismutase” OR Catalase OR “Glutathione peroxidase” OR antiox *).

The search strategy was developed in tandem with the project supervisor. We ran a sensitivity analysis using additional keywords such as “superoxide”,”H_2_O_2_”, “hydrogenperoxide” or “redox stress” but these did not change the search results significantly, therefore the authors were satisfied that the search string used captured most variations in terminology for both the disease in question (OM), as well as including any nomenclature that would be associated with the oxidative stress pathway.

Studies that were not published in English were excluded via an automatic tool, and any studies that were: reviews/systematic reviews/meta-analyses, book chapters and non-peer-reviewed literature were manually excluded by reviewers.

### 4.2. Data Extraction

Two sets of two reviewers independently evaluated titles and abstracts of articles, to screen for exclusion criteria. A kappa score of agreement was calculated for each pair, and disagreements were resolved by a third-party judge. The kappa scores for each round and anonymous pair are presented below (Table 5).

A predefined data extraction sheet was used to distill information from the full texts of eligible studies.

### 4.3. Data Synthesis

Once suitable studies for inclusion were determined, the evidence from the studies were presented in two ways. Data from the final search was presented as extraction tables, as seen in the Appendix A. This included the study type, population, intervention and comparator, outcome measurement, observed effect, and mechanism of action. Additionally, a narrative component was included in the results section. This detailed some of the interventions that were mentioned in multiple studies and compared the results between each of these studies.

## 5. Conclusions

This scoping review has revealed that there is a wide range of publications that have explored OM in the context of oxidative stress to varying extents, dating back to 1992, indicating that there has been interest in this field for some time. There has also been a revelation about the breadth of data that can actually directly support our area of interest—the majority of clinical studies do not delve into the antioxidant mechanisms behind their action, so perhaps this could be a future direction for research. This is likely the case due to clinical studies relying on prior in vitro and in vivo studies to define the mechanisms of action of these pathways.

Regarding the in vivo studies, it is clear that reduction in ROS, MPO, and MDA and increase in SOD and GSH are associated with prevention or improvement in OM severity. However, the exact mechanism by which oxidative stress contributes to chemo- and radiotherapy-induced OM is still largely unclear.

In regard to in vitro studies, further exploration on mechanisms of ROS reduction may be considered in future studies, in addition to ROS level detection.

Our data support a clear scope for further investigations into the oxidative stress pathways in OM.

## Figures and Tables

**Figure 1 ijms-23-04863-f001:**
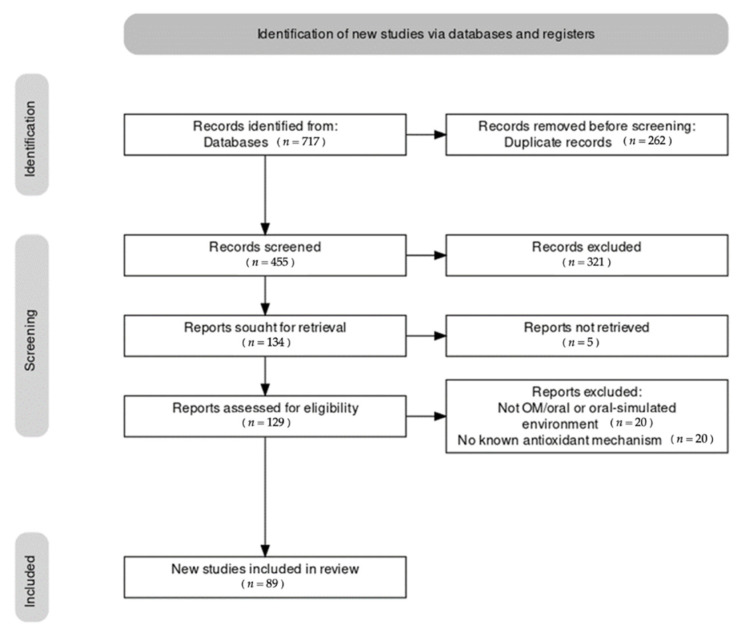
Results of the literature search. Note that during the identification stage, an indeterminate number of non-English articles were excluded prior to screening via automatic tool.

**Table 1 ijms-23-04863-t001:** Summary of in vitro study designs and results.

Author, Year	Population/Model	Intervention	Outcome/Effect Observed
Osaki et al., 1994 [6]	Sheep red blood cells	Azelastine	Azelastine suppressed neutrophil respiratory burst
Greenberger et al., 2014 [7]	Fancd2−/−, Fancd2+/−, and Fancd2+/+ mice (C57BL/6J mouse strain)	GS-nitroxide, JP4-039	Decreased production of hematopoietic cells
Yamaguchi et al., 2019 [8]	Human dermal fibroblast cell	Astaxanthin (5 µM)	AST reduces damage from cisplatin (ROS generation).
Vaillancourt et al., 2021 [9]	Oral keratinocyte cell line	Green tea extract	Treatment led to reduced ROS production
Iglesias-Bartolome et al., 2012 [10]	Oral keratinocyte cell	Rapamycin	Rapamycin increased the protein levels of MnSOD, Decreased ROS after radiation
Baek et al., 2014 [11]	Human keratinocyte (HaCaT) cells	3-amino-3-(4-fluoro-phenyl)-1H-quinoline-2, 4-dione	KR22332 inhibits radiation-induced intracellular ROS generation
Chang et al., 2014 [12]	Human keratinocyte (HaCaT) cells	KRG	KRG inhibits radiation-induced intracellular ROS generation
Lambros et al., 2014 [13]	Human buccal keratinocytes	QYD, or an NAC-QYD mixture	NAC-QYD treatment led to cell integrity prevention and prevention of apoptosis.
Maiguma et al., 2009 [14]	Human epidermal keratinocytes, periodontal ligament fibroblasts	Amifostine and COX-1 Inhibitor	Amifostine has hydroxyl radical scavenging activity and weak superoxide Radical scavenging activity.
Shin et al., 2013 [15]	Human keratinocyte line HaCaT	Epicatechin	EC significantly reduced radiation-induced intracellular ROS generation
Tsubaki et al., 2018 [16]	Primary NHOK cells	Rebamipide in combination with 5-FU or cisplatin	Rebamipide-induced Bcl-2 and Bcl-xL expressions prevented, and decreased Bax and Bim expressions
Kim et al., 2020 [17]	Human immortalized keratinocytes, HaCaT cells	Pre-treatment with NAC then radiation with LINAC	Radiotherapy-induced ROS generation was reduced
Takano et al, 2014 [18]	Human oral keratinocyte cell	γ-tocotrienol	γ-tocotrienol could prevent 5-FU-induced ROS generation
Cirillo et al., 2015 [19]	NHOF, Keratinocytes	H_2_O_2_ or TGF-β, with or without Mucosamin	Mucosamin showed a protective effect on oxidative stress-induced senescence.
Rupel et al., 2018 [20]	Human keratinocytes (HaCaT)	LPS followed by photomodulation	Significant ROS reduction.
Chung et al., 2009 [21]	Human embryonic fibroblasts	1 mM phenylbutyrate	Phenylbutyrate induced histone H3 hyperacetylation
Huth et al., 2020 [22]	Human dermal fibroblasts, epidermal keratinocytes, oral mucosa Keratinocytes/fibroblasts	Expanthenol-containing ointment	Reduction in ROS production.
Yoshida et al., 2014 [23]	Sa3 cells (gingival cells)	5-FU and TJ84 (Daiokanzoto)	TJ-84 reduced the chemotherapy-induced mitochondria-derived-O_2_.
Shinde et al., 2016 [24]	TC-1 epithelial cell from Fancd2+/+, Fancd2+/− and Fancd2−/− mice	JP4-039 before or after irradiation	Mitochondrial targeting makes F15/JP4-039 an effective protector from radiation.
Kim et al., 2017 [25]	Human pharyngeal cell	N/A	Decreased Intracellular ROS and increased DPPH-scavenging activity
Park et al., 2018 [26]	Human pharyngeal cell line	OCE treatment	OCE decreases intracellular ROS production

5-FU = 5 Fluorouracil; AST = Astaxanthin COX-1 = Cyclooxygenase-1; DPPH = 2,2-Diphenyl-1-Picrylhydrazyl; EC = Epicatechin; JP4-039 = Mitochondrial—Targeted Gramicin S Nitroxide; KR22332 = 3-Amino-3-(4-Fluorophenyl)-1H-Quinoline-2,4-Dione; KRG = Korean red ginseng; LPS = Lipopolysaccharide; MnSOD = Manganese Superoxide Dismutase; NHOK = Normal human oral keratinocytes; NAC = N-Acetyl Cysteine; NAC-QYD = N-Acetyl Cysteine and Qingre Liyan Decoction; NHOF = Normal human oral fibroblasts; OCE = Onchungeum; ROS = Reactive oxygen species.

**Table 2 ijms-23-04863-t002:** Summary of in vivo studies and main results.

Author, Year	Population/Model	Intervention	Outcome/Effect Observed
Greenberger et al., 2014 [7]	6–8-week-old Fanconi anaemia (Fancd2−/−) mice	JP4-039 with irradiation	Significant reduction in OM in treatment group.
Iglesias-Bartolome et al., 2012 [10]	Radiotherapy-induced OM rat model	Rapamycin	Reduction in p53, γH2AX expression and increase in SOD levels in treatment group associated with mucosal protection.
Shin et al., 2013 [15]	32 female Sprague Dawley rats (6 weeks old)	Epicatechin	Reduced intracellular ROS generation and prevented cell apoptosis
Kim et al., 2020 [17]	60 female Sprague Dawley rats (6 weeks old)	N-acetylcysteine	Reduced ROS production and preventive effect against OM
Shinde et al., 2016 [24]	Adult mice (C57BL/6) of 10 to 12 weeks of age (Fancd2+/+, Fancd2+/− and Fancd2−/−	F15/JP4-039 nitroxide or F15/4-aminoTempo with radiotherapy	Improvement from OM only in JP4-039 treatment group.
Kim et al., 2017 [25]	Male golden Syrian hamsters (7 weeks old)	S. militiorrhiza	Improved OM healing, reduced caspase-3, TNF-α and IL-1β, and nuclear NF-κB expression
Park et al., 2018 [26]	36 male golden Syrian hamsters (7 weeks old)	Onchung-eum	Improved oral mucositis recovery, and reduced ROS production
Ara et al., 2008 [27]	Male golden Syrian hamsters (5–6 weeks old)	Velafermin	Significantly reduced degree of OM and IL-6, but increased NF-E2-related factor-2 levels in treatment group
Cléemenson et al., 2019 [28]	Female C57BL/6 mice	CPh1014	Significantly reduced DNA damage and severity of radiation-induced OM in treatment groups.
Nakajima et al., 2015 [29]	Radiotherapy-induced OM mice model	Edaravone	Significantly reduced OM score, MPO, LPO and cell apoptosis in treatment groups.
Mafra et al., 2019 [30]	Chemotherapy-induced OM hamster model	Gliclazide	Significantly reduced OM score, MDA, MPO, TNF-α and IL-1β levels in treatment groups.
Yang et al., 2018 [31]	Radiotherapy-induced OM mice model	DMSO	Reduced ulcer size and increased mucosal thickness in treatment group.
Ortiz et al., 2015 [32]	Male Wistar rats	3% melatonin gel	Reduced severity of OM ulcers; reduced LPO, GPx, GRdand GSH/GSS
Shimamura et al., 2019 [33]	Chemotherapy-induced OM mice model	GGsTop	Significantly reduced ulcer area and improved healing of OM ulcers in treatment group.
Im et al., 2019 [34]	Female C57Bl/6 mice (5–8 weeks old)	NecroX-7	Reduced oxidative stress and cell apoptosis
Ala et al., 2020 [35]	33 Wistar rats	Sumatriptan with radiotherapy.	Significantly reduced MDA and TNF-α level, and epithelial thickness and mucosal damage in treatment groups.
Yoshino et al., 2014 [36]	Chemotherapy-induced OM hamster model	10% acetic acid with 5-FU or 5-FU alone or acetic acid alone	Significantly higher MDA level, associated reduced ROS decay and greater OM manifestation in acetic acid and combined intervention groups.
Vilar et al., 2020 [37]	Male golden Syrian hamsters	Gold nanoparticles with polyvinylpyrrolidone	Reduced IL-1β and increased GSH levels. Prevention of OM.
Gümüş et al., 2020 [38]	36 female Wistar albino rats (21–30 days old)	Topical TA or CHX with 5-FU	Significantly higher GSH and lower LPO level in treatment groups.
Takeuchi et al., 2018 [39]	20 ICR male mice (10 weeks old)	Rebamipide-loaded PLGA nanoparticles coated with chitosan	Reduced ulcer area
Nakashima et al., 2014 [40]	107 male Crl:CD Sprague Dawley rats (7–9 weeks old)	Rebamipide liquid with micro-crystals or submicronized crystals	Dose-dependent reduction of oral ulcer area and incidence
Nakashima et al., 2017 [41]	208 Crl:CD Sprague Dawley rats (6 weeks old)	Rebamipide liquid	Reduced tongue damaged area and inflammatory protein/gene expression
Koohi-Hosseinabadi et al., 2015 [42]	56 male golden hamsters (6–8 weeks old)	Z. jujuba hydroalcoholic extract	Reduced oral mucositis severity and MDA concentration, but increased SOD activity
Koohi-Hosseinabadi et al., 2017 [43]	56 male golden hamster	E. angustifolia hydroalcoholic extract	Reduced MDA and MPO levels, and increased SOD activity
Tanideh et al., 2019 [44]	90 male golden Syrian hamsters (8–10 weeks old)	T. ammi or P. atlantica extract	Increased oral epithelium density; reduced MPO and IL-1β levels; increased SOD activity
Watanabe et al., 2013 [45]	Male golden Syrian hamsters (7 weeks old)	Royal jelly	Reduced MPO activity, IL-1β and TNF-α expression, inflammatory cells and OM ulceration
Takuma et al., 2008 [46]	Male golden Syrian hamsters (6 weeks old)	E. japonica seed extract	Reduced plasma lipid peroxide level and mucositis severity
Rezvani & Ross, 2004 [47]	Female Sprague Dawley rats (12 weeks old)	Sunflower and α-tocopherol mix	Increased radiation threshold for OM onset
Gupta et al., 2020 [48]	48 C3H female mice (10 weeks old)	L. reuteri	Reduced in epithelial damage and oxidative stress
Cuba et al., 2020 [49]	Ninety CF-1 male mice (10 weeks old)	Cannabidiol	Reduced OM severity, inflammation, glutathione and catalase
Aghel et al., 2014 [50]	28 male Wistar rats (7–11 weeks old)	Propolis	Reduced mucositis severity and antioxidants
Motallebnejad et al., 2020 [51]	28 male Wistar albino rats (2.5–3 months old)	Lycopene	Lowered mean OM grade, reduced oxidative stress
Birer et al., 2017 [52]	Female C57BL/6 mice (6–8 weeks old)	MnBuOE	Increased GSH/GSSG ratio and reduced oxidative stress
Ashcraft et al., 2015 [53]	C57BL/6 mice	MnBuOE	Reduced OM incidence
Cruz et al., 2015 [54]	Female Syrian golden hamsters	Antimicrobial laser and aPDT	Most healing and less persistence of OM in treatment groups.
Thieme et al., 2020 [55]	Chemotherapy-induced OM rat model	Laser irradiation on both extra-oral and intra-oral	Reduction in OM score with associated glutathione peroxidase activity increase in extra-oral laser irradiation
Shen et al., 2018 [56]	Male C57BL/6 mice	CXCR2 -overexpressing MSCswith radiotherapy	Significant reduction in ROS production and accelerated healing of OM in treatment group.
Willis et al., 2018 [57]	Double-knockout mice, Fancg−/−, Fanca−/−, Fancd2−/−	Mitochondrial-targeted JP4- 039/F15	Reduced OM coverage and TGF-β mRNA elevation in treatment group.
Cortrim et al., 2012 [58]	Radiotherapy and chemotherapy-induced OM rat model	Tempol	Reduced ulceration size in both radiotherapy and chemotherapy-induced model treatment group.
Hu et al., 2017 [59]	Radiotherapy-induced OM miniature pig model	Tempol	Lower OM scores, area of ulcers, and higher buccal epithelial thickness in treatment groups.
Üçüncü et al., 2006 [60]	Radiotherapy-induced OM rat model	Vitamin E or L-carnitine or Mix	Later onset of OM and inflammation, decreased MDA, and increased SOD and CAT levels in treatment groups.
Luo et al., 2019 [61]	Radiotherapy-induced OM female mice model with human oral tissue xenotransplant	Tat-Smad7	Reduction in size of ulceration, DNA damage, cell apoptosis and inflammatory infiltration, and increased epithelial proliferation
Guo et al., 2003 [62]	C3H/HeNsd mice	Plasmid DNA of human SOD2 transgene with radiotherapy	Significant increase in SOD2 level and decrease in ulceration in treatment groups.
Epperly et al., 2004 [63]	Radiotherapy-induced OM rat model	MnSOD plasmid	Reduced ulceration and apoptotic cells in treatment groups.
Tao et al., 2019 [64]	Radiotherapy-induced OM mice model	25Gy irradiation	miR-200, TNF-α, MIP-1β, and IL-1α expression in irradiation group increased.

aPDT = photodynamic therapy; CHX = Chlorhexidine; CPh-1014 = Free radical scavenger; CXCR2 = CXC chemokine receptor 2; GGsTop = Irreversible γ-glutamyl transpeptidase inhibitor; GPx = Glutathione peroxidase; GRd = Glutathione reductase; GSH/GSSG = Glutathione; IL-1α = Interleukin-1α; IL-1β = Interleukin-1β; IL-6 = Interleukin-6; JP4-039/F15 = GS-nitroxide; LPO = Lipid peroxidation; MDA: Malondialdehyde; MnSOD = Manganese superoxide dismutase; MPO = Myeloperoxidase; MSCs = Mesenchymal stem cells; NecroX-7 = Tetrahydropyran-4-yl; PLGA = Poly (lactic-co-glycolic acid); SOD = Superoxide dismutase; TA = Triamcinolone acetonide; Tat-Smad7 = Smad7 protein with Tat tag that permeates cells; TGF-β = Transforming growth factor β; TNF-α = Tumour necrosis factor α.

**Table 3 ijms-23-04863-t003:** Summary table of results of clinical studies.

Author, Year	Population	Intervention	Outcome/Effect Observed
Osaki et al., 1994 [6]	CT pts.	Azelastine + Vitamin C + Vitamin E + Glutathione	Delayed and less severe inflammation in Azelastine intervention.
Cirillo et al., 2015 [19]	Pts. receiving radio- and/or CT	Mucosamine	Prevention or reduced grade of OM
Rupel et al., 2018 [20]	Pts. affected by grade 2 or 3 OM aged 40–95 years, diagnosed with solid or hematologic malignancy undergoing CT and/or RT, and available to undergo PBM for 4 consecutive days	Photobiomodulation	660 nm laser light increases ROS, whereas the 970 nm light exerted a moderate antioxidant activity. The 800 nm light or the combination of the 3 wavelengths resulted in the largest ROS reduction.
Sung et al., 2007 [65]	Children aged 6–17 years old undergoing CT	Vitamin E	No significant difference in objective mucositis scores
Ferreira et al., 2004 [66]	Pts. with oral cavity and oropharynx cancer	Vitamin E	Vitamin E reduced pain grades.
El- Housseiny et al., 2007 [67]	Cancer pts. receiving CT	Vitamin E	Vitamin E resulted in complete resolution of OM in 80% of pts.
Wadleigh et al., 1992 [68]	Pts. receiving CT for malignancy	Vitamin E	Vitamin E resulted in relatively more instances of complete resolution of OM.
Sayed et al., 2019 [69]	HNC pts. receiving 30–35 RT fractions with or without CT	Pentoxifylline and vitamin E	Pentoxifylline and vitamin E did not affect the incidence of OM, however, after adjusting for age, it decreased the incidence of severe OM and decreased the duration of OM
Agha- Hosseini et al., 2021 [70]	Pts. with H&N cancer undergoing RT	Vitamin E + triamcinolone + HA	Reduction in OM grade and pain intensity
Khurana et al., 2013 [71]	Children with CT-induced OM	Vitamin E or Pycnogenol	Both interventions equally effective at reducing OM severity and pain.
Venkatesh et al., 2014 [72]	HNC pts. undergoing CRT or RT	None- observational study of association of genetic polymorphism with OM	NBN variants are of predictive significance in analysing oral mucositis prior to radiotherapy.
Pratesi et al., 2011 [73]	SCCHN pts. following RT	None—observational study of genetic polymorphisms	XRCC1-399Gln allele significantly associated with higher risk of OM, with increased incidence of higher grades.
Urbain et al., 2012 [74]	Adults treated with alloHCT with CT	None—prospective study of AOX concentrations relative to OM severity	No single AOX had predictive value for severity or incidence of OM.
Wardman et al., 2013 [75]	HNC treated with CHART	None—observational study of correlation of plasma AOX and OM severity	No correlation between mucositis severity and measures of plasma AOX.
Bachmeier et al., 2014 [76]	BMT pts.	None- observational study measuring changes in AOX activity	Post BMT, 85% developed OM, increase in SOD and decrease in UA during M-stage
Severin et al., 2005 [77]	Leukemia pts. receiving BMT and healthy blood donors	None- observational study of AOX capacity before and after radiation	Severe OM associated with specific depletion in leukocyte, lymphocyte and plasma antioxidant concentration
Chaitanya et al., 2017 [78]	CRT pts.	Rebamipide	Decreased OM severity and pain intensity, delayed onset
Ishii et al., 2017 [79]	Stomatitis pts. with CT	Rebamipide	Rebamipide pts experienced improvements or eliminations of SES score.
Gholizadeh et al., 2017 [80]	Pts. with AML undergoing CT	Zinc sulfate	Frequency of severe OM reduced
Doi et al., 2015 [81]	Newly diagnosed pts. with HNC and undergoing RT	Polaprezinc	Decreased incidence of grade 3 OM, promoted recovery
Büntzel et al., 2010 [82]	RT pts.	Selenium	No significant difference was observed.
Jahangard-Rafsanjani et al., 2013 [83]	Leukaemia pts. undergoing HSCT with HDC	Selenium	Selenium had an effect on incidence of severe OM (reduced), and its duration (reduced). Glu.Px increased in intervention arm.
Bardellini et al., 2016 [84]	Paediatric pts. receiving CT for ALL	Mucosyte	Intervention demonstrated declines in OM and pain.
Tacyildiz et al., 2010 [85]	Children undergoing CT + RT	Genistein	Less OM occurred with genistein treatment.
Wu et al., 2010 [86]	Nasopharyngeal cancer pts. receiving CRT + induction CT	Actovegin	Actovegin group had fewer incidences of grade 2, 3 OM, and reduced patient pain grading in preventive arm.
Anderson et al., 2019 [87]	Pts. with locally advanced oral or oropharyngeal cancer	GC4419	Dose of 30 mg produced intermediate improvements, and at a dose of 90 mg produced reduced duration, incidence and severity of severe OM.
Yokomizo et al., 2004 [88]	Pts. with advanced or recurrent colon cancer	Allopurinol ice	Decreased incidence and severity
Hosseinjani et al., 2017 [89]	Adults with non-Hodgkin’s lymphoma, Hodgkin disease or multiple myeloma undergoing autologous HSCT	Erythropoietin	Significant reduction in the incidence and duration of OM
Vidal-Casariego et al., 2013 [90]	Pts. treated with RT for H&N or cancer in chest area	Glutamine	Decreased risk of developing OM and ARIE
Moslehi et al., 2014 [91]	Pts. with AML, ALL or MDS undergoing HSCT with HDC	NAC	Duration and frequency (severe only) of OM reduced in NAC group. Glu.Px increased in intervention arm.
Salehi et al., 2018 [92]	Colon cancer pts.	Propolis	Propolis group had significantly reduced severity of OM relative to placebo.
Naidu et al., 2005 [93]	CRT induced OM pts	MF5232	OM grade improves with intervention.
Elsabagh et al., 2019 [94]	HNC pts. receiving RT	Melatonin	Discomfort/pain lower in intervention arm. TAC reduced in control group.
Elkerm et al., 2014 [95]	Pts. with H&N cancer prior to exposure to first-line treatment	Date palm pollen	Significantly reduced OMAS score and pain severity of OM
Mills, 1998 [96]	Pts. with SCC undergoing CT	Beta-carotene supplementation	Intervention arm developed severe disease later and to a lesser extent.
Babaee et al., 2013 [97]	H&N cancer pts.	*Calendula* officinalis flower	Lower OMAS

ALL = acute lymphoblastic leukemia; alloHCT = allogenic hematopoietic cell transfer; AML = acute myeloblastic leukemia; AOX = antioxidant; ARIE = Acute radiation-induced esophagitis; BMT = Bone marrow transplant; CHART = Continuous Hyperfractionated Accelerated RadioTherapy; CRT = chemoradiation; CT = chemotherapy; HA = hyaluronic acid; H&N = Head and Neck; HDC = high-dose chemotherapy; HNC = head and neck cancer; HSCT = hematopoietic stem cell transfer; MDS= myelodysplastic syndrome; NAC = N-acetyl cysteine; OMAS = Oral mucositis assessment scale; OM = oral mucositis; PBM = photobiomodulation; Pts. = patients; RT= radiotherapy; SCC = squamous cell carcinoma; SCCHN = squamous cell carcinoma of the head and neck; SES = stomatitis evaluation score; SOD = superoxide dismutase; TAC = total antioxidant capacity; UA = uric acid.

**Table 4 ijms-23-04863-t004:** Interventions with statistically significant results for various outcomes.

OUTCOME	STUDY
**Reduced incidence of OM**	Genistein [85], Actovegin * [86], GC4419 [87], Allopurinol [88], Erythropoietin [89], Glutamine [90]
**Reduced severity of OM**	GC-4419 [87], Allopurinol [88], NAC [91], Propolis [92], MF5232 [93], Melatonin [94], Date Palm Pollen [95], Beta-carotene [96], *Calendula* [97]
**Reduced duration of OM**	GC4419 [87], NAC [91], Erythropoietin [98]
**Reduced pain associated with OM**	Melatonin [94], Date palm pollen [95]

* Actovegin only reduced the incidence of Grade 2 and Grade 3 OM to statistical significance.

**Table 5 ijms-23-04863-t005:** Kappa scores for inclusion and exclusion.

Round	Pair	Kappa
Exclusion by Study Type	1	0.90
	2	0.70
Exclusion by Title and Abstract	1	0.86
	2	0.75

## Data Availability

All datasets are available upon reasonable request to the corresponding author.

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
