# Peer review of "Oxidative Stress and Chemoradiation-Induced Oral Mucositis: A Scoping Review of In Vitro, In Vivo and Clinical Studies"

_ijms, 2022, doi:10.3390/ijms23094863_

Round 1

Reviewer 1 Report

The authors with their review article "Oxidative stress and chemoradiation induced oral mucositis: a scoping review” gives a nice and clear overview about the literature dealing with oxidative stress in oral mucositis induced by chemoradiation.

However the following points should be cover:

  • It is not clearly stated how oxidative stress is defined. Please give a short introduction about oxidative stress and whether this definition influenced the literature search.
  • To identify relevant publication several keywords associated with “oxidative stress” were used. However some keywords are missing which are clearly associated with “oxidative stress” such as “superoxide”,”H2O2”, “hydrogenperoxide” or also “redox stress”. At least it should be explained why these keywords were not included in the literature search.

Reviewer 2 Report

The Review by Nguyen et al is well conduct and deeply explored the role of oxidative stress in the development of Oral Mucositis a debilitating condition of the gastrointestinal tract which may appare after antineoplastic treatment.

However, three recent in vitro studies (published between 2020 and 2022) explored the role of oxidative stress in oral mucositis and were not included.

1)Picciolo G, Mannino F, Irrera N, Minutoli L, Altavilla D, Vaccaro M, Oteri G, Squadrito F, Pallio G. Reduction of oxidative stress blunts the NLRP3 inflammatory cascade in LPS stimulated human gingival fibroblasts and oral mucosal epithelial cells. Biomed Pharmacother. 2022 Feb;146:112525. doi: 10.1016/j.biopha.2021.112525.

2)Picciolo G, Mannino F, Irrera N, Altavilla D, Minutoli L, Vaccaro M, Arcoraci V, Squadrito V, Picciolo G, Squadrito F, Pallio G. PDRN, a natural bioactive compound, blunts inflammation and positively reprograms healing genes in an "in vitro" model of oral mucositis. Biomed Pharmacother. 2021 Jun;138:111538. doi: 10.1016/j.biopha.2021.111538. 

3)Picciolo G, Pallio G, Altavilla D, Vaccaro M, Oteri G, Irrera N, Squadrito F. β-Caryophyllene Reduces the Inflammatory Phenotype of Periodontal Cells by Targeting CB2 Receptors. Biomedicines. 2020 Jun 17;8(6):164. doi: 10.3390/biomedicines8060164.

Authors should add these articles for completeness of information and to improve the quality of the paper.

Round 2

Reviewer 1 Report

The authors have answered all my questions.